# *HOUND*: High-Order Universal Numerical Differentiator for a Parameter-free Polynomial Online Approximation

**Igor Katrichek**
katrichek@gmail.com

## Abstract

This paper introduces a scalar numerical differentiator, represented as a system of nonlinear differential equations of any high order. We derive the explicit solution for this system and demonstrate that, with a suitable choice of differentiator order, the error converges to zero for polynomial signals with additive white noise. In more general signal cases, the error remains bounded, provided that the highest estimated derivative is also bounded. A notable advantage of this numerical differentiation method is that it does not require knowledge of the Lipschitz constant value and does not require tuning parameters based on the specific characteristics of the signal being differentiated. We propose a discretization method for the equations that implements a cumulative smoothing algorithm for time series. This algorithm operates online, without the need for data accumulation, and it solves both interpolation and extrapolation problems without fitting any coefficients to the data.

## 1 Introduction

Numerical differentiation is a common challenge in digital signal processing and automatic control system design. The difficulty of this task increases with the order of the derivative being estimated, and even more so when the signal contains noise. Existing methods typically require prior knowledge of the signal and noise characteristics, such as the Lipschitz constant (bound value of the highest estimated derivative) and noise variance. Differentiator parameters are then tuned based on this information for specific signals or signal classes. Most numerical differentiators are designed for known orders, such as first or second derivatives, and are implemented as recurrence algorithm and formulas. A comprehensive review of such methods can be found in the literature Mojallizadeh et al. (2021).

The online estimation of all derivatives up to an arbitrarily chosen high order can be achieved using both the super-twisting method (Subsection 2.3 in Mojallizadeh et al. (2021)) and the high-gain method (Subsection 2.9 in Mojallizadeh et al. (2021)). However, the optimal coefficients for these approaches have been determined only for derivatives of bounded order (see Table 2 in Mojallizadeh et al. (2021)). These methods have been established in the literature for quite some time, and subsequent research in online high-order numerical differentiation has focused on optimal tuning and discretization strategies, as well as on refining the convergence region and enhancing stability. Moreover, it should be noted that, for signals corrupted by noise, all known methods yield an asymptotically non-zero differentiation error, which is at most bounded by a constant that depends on the noise level.

The goal of this paper is to develop a numerical differentiation method that eliminates the need for tuning. The proposed method is capable of computing all derivatives up to the highest bounded derivative, even for noisy signals. The differentiation results are then applied to automatically perform polynomial regression on the differentiable signal.

## 2 FORMULATION OF RESULTS

We define the following notation: $t \in \mathbb{R}_{>0}$ is the independent variable (for signals, this is usually time), $f(t) \in \mathbb{R}$ is the signal that is differentiable at least $n$ times, $z_0(t) \in \mathbb{R}$ is the estimate of the signal, $z_m(t) \in \mathbb{R}$ is the estimate of the $m$-th derivative of the signal with respect to $t$. Given $n = 1, 2, 3, \ldots$, and $m = 1, 2, \ldots, n$, the High-Order Universal Numerical Differentiator (HOUND) is represented by the following system of continuous differential equations:

$$z_{m-1}^{(1)}(t) = z_m(t) - \frac{(n+m-1)!}{m!(n-m)!} \frac{n}{t^m} (z_0(t) - f(t)), \tag{1}$$

Here, $(n-1)$ represents the order of the differentiator ($n = 1$ estimates the signal itself, $n = 2$ estimates its first derivative, and so on), with $z_n(t) \equiv 0$. Throughout the paper, $y^{(d)}(t)$ denotes the $d$-th derivative of $y$ with respect to $t$. Specifically, $z_{m-1}^{(1)}(t)$ represents the first derivative estimate of the $(m-1)$-th derivative of the signal.

In essence, the differentiator in Equation (1) is a nonlinear version of the Luenberger observer (which is essentially the same as a high-gain observer from Vasiljevic & Khalil (2008)) applied to a chain of integrators. The key difference is that the "constant" preceding the observation error is not tuned but predefined, and it decreases inversely with a $m$-th power of $t$.

For $n = 1, 2, 3$, the differentiator from Equation (1) is of the form (we omit the explicit dependence on $t$ for brevity):

$$z_0^{(1)} = -\frac{1}{t}(z_0 - f) \qquad \text{if } n = 1,$$

$$\begin{cases} z_0^{(1)} = -\frac{4}{t}(z_0 - f) + z_1 \\ z_1^{(1)} = -\frac{6}{t^2}(z_0 - f) \end{cases} \qquad \text{if } n = 2,$$

$$\begin{cases} z_0^{(1)} = -\frac{9}{t}(z_0 - f) + z_1 \\ z_1^{(1)} = -\frac{36}{t^2}(z_0 - f) + z_2 \\ z_2^{(1)} = -\frac{60}{t^3}(z_0 - f) \end{cases} \qquad \text{if } n = 3.$$

In the Appendix A, we provide the derivation of the solution to the system (1):

$$z_{m-1}(t) = f^{(m-1)}(t) + \sum_{d=1}^{n} \frac{a_{m,d,n}}{t^{d+m-1}} \left( c_d + \frac{(-1)^d}{b_{d,n}} \int_{t_0}^{t} \tau^{d+n-1} f^{(n)}(\tau) \, d\tau \right), \tag{2}$$

where $a_{m,d,n}$ and $b_{d,n}$ are non-zero integer constants, determined by the respective indices and the order of the differentiator.

The Appendix A also demonstrates that if the highest estimated derivative of a signal is bounded by the Lipschitz constant $L$ (i.e., $|f^{(n-1)}(t)| \leq L$), then estimation errors are bounded by the following expression: $|z_{m-1}(t) - f^{(m-1)}(t)| \leq 2L + \sum_{d=1}^{n} \frac{a_{m,d,n}}{t^{d+m-1}} \left( |c_d| + \frac{t_0^{d+n-1} 2L}{b_{d,n}} \right)$ which tends to $2L$.

Here are examples of Equation (2) for $n = 1, 2, 3$:

$$z_0(t) = f(t) + \frac{1}{t}(c_1 - \int_{t_0}^{t} \tau f^{(1)}(\tau) \, d\tau) \qquad \text{if } n = 1,$$

$$\begin{cases} z_0(t) = f(t) \quad + \frac{1}{t}(c_1 - \int_{t_0}^{t} \tau^2 f^{(2)}(\tau) \, d\tau) + \frac{1}{t^2}(c_2 + \int_{t_0}^{t} \tau^3 f^{(2)}(\tau) \, d\tau) \\ z_1(t) = f^{(1)}(t) + \frac{3}{t^2}(c_1 - \int_{t_0}^{t} \tau^2 f^{(2)}(\tau) \, d\tau) + \frac{2}{t^3}(c_2 + \int_{t_0}^{t} \tau^3 f^{(2)}(\tau) \, d\tau) \end{cases} \quad \text{if } n = 2,$$

$$\begin{cases} z_0(t) = f(t) \quad + \frac{1}{t}(c_1 - \frac{1}{2}\int_{t_0}^{t} \tau^3 f^{(3)}(\tau) \, d\tau) + \frac{1}{t^2}(c_2 + \int_{t_0}^{t} \tau^4 f^{(3)}(\tau) \, d\tau) + \frac{1}{t^3}(c_3 - \frac{1}{2}\int_{t_0}^{t} \tau^5 f^{(3)}(\tau) \, d\tau) \\ z_1(t) = f^{(1)}(t) + \frac{8}{t^2}(c_1 - \frac{1}{2}\int_{t_0}^{t} \tau^3 f^{(3)}(\tau) \, d\tau) + \frac{7}{t^3}(c_2 + \int_{t_0}^{t} \tau^4 f^{(3)}(\tau) \, d\tau) + \frac{6}{t^4}(c_3 - \frac{1}{2}\int_{t_0}^{t} \tau^5 f^{(3)}(\tau) \, d\tau) \\ z_2(t) = f^{(2)}(t) + \frac{20}{t^3}(c_1 - \frac{1}{2}\int_{t_0}^{t} \tau^3 f^{(3)}(\tau) \, d\tau) + \frac{15}{t^4}(c_2 + \int_{t_0}^{t} \tau^4 f^{(3)}(\tau) \, d\tau) + \frac{12}{t^5}(c_3 - \frac{1}{2}\int_{t_0}^{t} \tau^5 f^{(3)}(\tau) \, d\tau) \\ \text{if } n = 3 \end{cases}$$

For an unnoised polynomial signal of order $(n-1)$, expressed as $f(t) = \sum_{j=0}^{n-1} K_j t^j$ (where $K_j$ are unknown real constants), as $t$ increases, the estimates of the signal's derivatives converge to their true values. Specifically, the signal estimate converges to the signal itself. This means that $\lim_{t\to\infty} z_{m-1}(t) = f^{(m-1)}(t)$. This occurs because the higher derivatives of the polynomial signal, starting from the $n$-th order, are zero: $f^{(n)}(t) = 0$. The signal's derivatives can be computed using the formula $f^{(m-1)}(t) = \sum_{j=m-1}^{n-1} \frac{j!}{(j-m+1)!} K_j t^{j-m+1}$. By applying the Taylor formula, we can estimate the unknown constants at each time step:

$$K_j = \frac{f^{(j)}(0)}{j!} \approx \frac{1}{j!} \sum_{i=j}^{n-1} \frac{(-t)^{i-j}}{(i-j)!} z_i(t), \tag{3}$$

which play the role of parameters of the polynomial regression in supervised machine learning.

For an unnoised polynomial signal of order $n$, expressed as $f(t) = \sum_{j=0}^{n} K_j t^j$, where $K_n$ is a non-zero constant, the estimates of the signal and its derivatives do not converge to the true values as $t$ increases. Specifically, the estimate $z_{m-1}(t)$ grows as $O(t^{n-m+1})$, because the highest derivative of the signal is a non-zero constant $f^{(n)}(t) = n! K_n$. Therefore, it is essential to choose a differentiator order that is at least as high as the order of the polynomial signal.

If the order of the differentiator exceeds the order of the polynomial signal, the estimates of the higher derivatives of the signal will approach zero as $t$ increases. For instance, if a signal of order $N$ is given by $f(t) = \sum_{j=0}^{N} K_j t^j$, where $N < n-1$, then for $j > N$, $\lim_{t\to\infty} z_j(t) = 0$ because $f^{(j)}(t) = 0$. As a result, the estimates of the unknown constants $K_j = \frac{f^{(j)}(0)}{j!}$ at $j > N$ also tend to zero. From a machine learning perspective, this behavior resembles automatic regularization.

For a harmonic signal, which is a linear combination of harmonics with constant amplitudes, frequencies, and phases, it can be shown that as $t$ increases, the estimates of both the signal and its derivatives tend to zero ($\lim_{t\to\infty} z_{m-1}(t) = 0$). Consequently, when a signal composed of both polynomial and harmonic components is input into the differentiator, the estimates of the harmonic components will vanish over time. After enough periods have passed, only the estimates of the polynomial signal and its derivatives will remain.

For a signal corrupted by additive white Gaussian noise, expressed as $f(t) = f_0(t) + \eta_0(t)$, where the noise $\eta_0(t)$ follows a normal distribution $N(0, \sigma_0^2)$ with zero mean and bounded variance, the differentiator can be described by a system of stochastic differential equations. These equations hold under the same conditions for $n = 1, 2, 3, \ldots$ and $m = 1, 2, \ldots, n$, with $z_n(t) \equiv 0$:

$$dz_{m-1}(t) = z_m(t)dt - \frac{(n+m-1)!}{m!(n-m)!} \frac{n}{t^m} \Big( \big(z_0(t) - f_0(t)\big)dt - \sigma_0 dW_0(t) \Big), \tag{4}$$

where $W_0(t) \sim N(0, t)$ is a Wiener process. The mean of the solution to system (4) matches the solution (2) of system (1) for the unnoised signal $f_0(t)$. It can be shown that the variance of the estimate

$$Var\big(z_{m-1}(t)\big) = K_{m-1} \frac{\sigma_0^2}{t^{2m-1}}, \tag{5}$$

decreases as $t$ increases. This means that the noise $\eta_0$ is effectively filtered out, allowing the extraction of the unnoised signal $f_0$ and its derivatives. Here, $K_{m-1}$ are constants.

Additionally, if signal's derivatives are corrupted by additive white Gaussian noise, then the variance $Var\big(z_{m-1}(t)\big)$ grows as $O(t^{2d-2m-1} \sigma_{d-1}^2)$. As $t$ increases, the variance becomes unbounded for all derivative estimates except the highest order $d = n$, due to the accumulation of noise through the integration of Wiener processes.

The differentiator (Equation (1)) approximates the signal and its derivatives, both in interpolation and extrapolation, using the Taylor series expansion:

$$\hat{f}^{(m-1)}(\tau) = \sum_{k=m-1}^{n-1} \frac{z_k(t)}{(k-m+1)!} (\tau - t)^{k-m+1}, \tag{6}$$

where $\tau \in \mathbb{R}$ represents the model independent variable (typically model time for signals).

If the estimates $z_{m-1}(t) = f^{(m-1)}(t)$ for the signal and its derivatives are accurate, and the time step $\Delta t = t - t_{prev}$ is small, then $z_{m-1}(t) \approx \sum_{k=m-1}^{n-1} \frac{z_k(t_{prev})}{(k-m+1)!} (t - t_{prev})^{k-m+1}$ according to Taylor formula. This allows us to discretize the equations (1), following the approaches of Brown R.G. (1963) and Holt C.C. (2004):

$$\begin{cases} z_{m-1}[t] = \left( \sum_{k=m-1}^{n-1} \frac{z_k[t-\Delta t]}{(k-m+1)!} \Delta t^{k-m+1} \right) + \Delta t \frac{(n+m-1)!}{m!(n-m)!} \frac{n}{t^m} \varepsilon[t] \\ \varepsilon[t] = \left( f[t] - \sum_{k=0}^{n-1} \frac{z_k[t-\Delta t]}{k!} \Delta t^k \right) \end{cases} . \tag{7}$$

For instance, when $\Delta t = 1$ and $n = 1$ the system becomes:

$$\begin{cases} z_0[t] = z_0[t-1] + \frac{1}{t} \varepsilon[t] \\ \varepsilon[t] = f[t] - z_0[t-1] \end{cases} ,$$

When $\Delta t = 1$ and $n = 2$ the system becomes:

$$\begin{cases} z_0[t] = z_0[t-1] + z_1[t-1] + \frac{4}{t} \varepsilon[t] \\ z_1[t] = z_1[t-1] + \frac{6}{t^2} \varepsilon[t] \\ \varepsilon[t] = f[t] - (z_0[t-1] + z_1[t-1]) \end{cases} .$$

Let us rewrite these equations in the classical form:

$$z_0[t] = \frac{1}{t} f[t] + (1 - \frac{1}{t}) z_0[t-1] \qquad \text{if } n = 1, \tag{8}$$

$$\begin{cases} z_0[t] = \frac{4}{t} f[t] + (1 - \frac{4}{t})(z_0[t-1] + z_1[t-1]) \\ z_1[t] = \frac{3}{2t}(z_0[t] - z_0[t-1]) + (1 - \frac{3}{2t}) z_1[t-1] \end{cases} \quad \text{if } n = 2. \tag{9}$$

Equation (8) is equivalent to Brown's method, where the smoothing factor $\frac{1}{t}$ changes based on the number of calculation steps $t = 1, 2, 3, \ldots$ . This approach is often referred to as cumulative averaging. Similarly, the equations in (9) correspond to Holt's method (Hyndman & Athanasopoulos (2021)), where $\alpha = \frac{4}{t}$ and $\beta^* = \frac{3}{2t}$ decrease in proportion to the calculation steps. This is analogous to double exponential smoothing, so we can refer to the recurrent equation (9) as double cumulative smoothing. Recurrent equation (7) can be referred to as high-order cumulative smoothing algorithm. In this context, smoothing refers to approximating the data with a polynomial of a specified order of smoothness, while filtering out additive white Gaussian noise. Cumulativity implies that the approximation begins at the initial value $t_0$ and expands to the current value $t$ using a window that covers the entire data set with equal weights.

Pseudocode for high-order cumulative smoothing (7) is given in Algorithm 1. The time complexity of the online algorithm is constant $O(1)$ for each $f[t]$. So the computational complexity of the offline algorithm is linear relative to the size of the collected data set. It's reasonable to assume the following initial conditions: $z_0[t_0] = f[t_0]$ and $z_m[t_0] = 0$ if prior estimates of the signal's derivatives are unavailable. A key feature of the algorithm which sets it apart from traditional time series smoothing methods, is its ability to handle calculations at irregular (unevenly) time steps $\Delta t[t]$. This allows for computations to occur only when new signal samples are received, provided they differ in value from the previous ones. Pseudocode for polynomial approximation (6) is given in Algorithm 2.

From a machine learning perspective, Equation (1) can be seen as a one-dimensional gradient flow or gradient descent (for $n = 1$) with a dynamic, adaptive learning rate $\gamma(t) = \frac{(n+m-1)!}{m!(n-m)!} \frac{n}{t^m}$.

The mean squared error (MSE) loss function $F(z) = \frac{1}{2}(z_0(t) - f(t))^2$ has a derivative (gradient) $\frac{dF(z)}{dz} = z_0(t) - f(t)$. Therefore, Algorithm (7) operates as a parameter-free, one-dimensional optimization algorithm that minimizes error with sublinear, or more specifically, logarithmic, convergence. This is because, under the condition of a bounded highest estimated derivative, the estimation error $z_0(t) - f(t)$ according to (2) decreases inversely proportional to the calculation step ($t^{-1}$). For $n > 1$ other error terms decrease according to negative powers of $t$ (i.e. $t^{-d}$, where $d = 1, \ldots, n$).

---

**Algorithm 1** High-Order Cumulative Smoothing

1: **Input:** Scalar data sequence $f[t]$ for $t = t_0 + \Delta t, t_0 + 2\Delta t, t_0 + 3\Delta t, \ldots$, scalar variables $z_{m-1}$ for $m = 1, 2, \ldots, n$ with initial values $z_0[t_0] = f[t_0]$ and $z_m[t_0] = 0$ where $t_0 > 0$
2: **Output:** Updated variables values $z_{m-1}[t]$ containing estimates of the signal and its derivatives
3: $t \leftarrow t_0$
4: **for** $i = 1, 2, 3, \ldots$ **do**
5: $\quad t \leftarrow t + \Delta t$
6: $\quad \varepsilon[t] \leftarrow f[t] - \sum_{k=0}^{n-1} \frac{z_k[t-\Delta t]}{k!} \Delta t^k$
7: $\quad$ **for** $m = 1, 2, \ldots, n$ **do**
8: $\quad\quad z_{m-1}[t] \leftarrow \left( \sum_{k=m-1}^{n-1} \frac{z_k[t-\Delta t]}{(k-m+1)!} \Delta t^{k-m+1} \right) + \Delta t \frac{(n+m-1)!}{m!(n-m)!} \frac{n}{t^m} \varepsilon[t]$
9: $\quad$ **end for**
10: **end for**

---

**Algorithm 2** Polynomial Approximation

1: **Input:** Updated scalar variables values $z_{m-1}[t]$ at the $t$ for $m = 1, 2, \ldots, n$ from Algorithm 1
2: **Output:** Polynomial approximated values $\hat{f}^{(m-1)}[\tau]$ of the signal and its derivatives at the $\tau$, coefficients $K_{m-1}$ values of the approximating polynomial $\hat{f}[\tau] = \sum_{m=1}^{n} K_{m-1} \tau^{m-1}$
3: **for** $m = 1, 2, \ldots, n$ **do**
4: $\quad \hat{f}^{(m-1)}[\tau] := \sum_{k=m-1}^{n-1} \frac{z_k[t]}{(k-m+1)!} (\tau - t)^{k-m+1}$
5: $\quad K_{m-1} := \frac{1}{(m-1)!} \sum_{i=m-1}^{n-1} \frac{z_i[t]}{(i-m+1)!} (-t)^{i-m+1}$
6: **end for**

---

## 3 DEMONSTRATION OF RESULTS

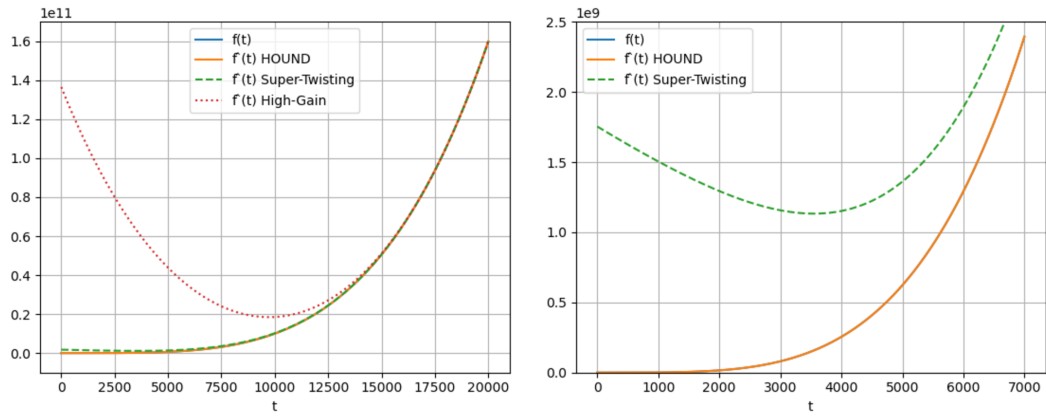

Figure 1: The differentiable signal $f(t)$ and its interpolation $\hat{f}(t)$

To demonstrate the differentiator's performance, consider a noisy polynomial signal of order 4: $f(t) = f_0(t) + \eta_0(t) = (5 - 0.004t + 0.0003t^2 - 0.00002t^3 + 0.000001t^4) + N(0, 0.7^2)$, over the range $t = 0$ to $20000$. For this signal, the corresponding differentiator with $n = 5$ and the recurrence Algorithm (7) for cumulative smoothing is expressed as:

$$\begin{cases} z_0[t] = \frac{25\Delta t}{t}\varepsilon[t] + z_0[t-\Delta t] + z_1[t-\Delta t]\Delta t + z_2[t-\Delta t]\frac{\Delta t^2}{2} + z_3[t-\Delta t]\frac{\Delta t^3}{6} + z_4[t-\Delta t]\frac{\Delta t^4}{24} \\ z_1[t] = \frac{300\Delta t}{t^2}\varepsilon[t] + z_1[t-\Delta t] + z_2[t-\Delta t]\Delta t + z_3[t-\Delta t]\frac{\Delta t^2}{2} + z_4[t-\Delta t]\frac{\Delta t^3}{6} \\ z_2[t] = \frac{2100\Delta t}{t^3}\varepsilon[t] + z_2[t-\Delta t] + z_3[t-\Delta t]\Delta t + z_4[t-\Delta t]\frac{\Delta t^2}{2} \\ z_3[t] = \frac{8400\Delta t}{t^4}\varepsilon[t] + z_3[t-\Delta t] + z_4[t-\Delta t]\Delta t \\ z_4[t] = \frac{15120\Delta t}{t^5}\varepsilon[t] + z_4[t-\Delta t] \\ \varepsilon[t] = f[t] \quad - \quad \left( z_0[t-\Delta t] + z_1[t-\Delta t]\Delta t + z_2[t-\Delta t]\frac{\Delta t^2}{2} + z_3[t-\Delta t]\frac{\Delta t^3}{6} + z_4[t-\Delta t]\frac{\Delta t^4}{24} \right) \end{cases}$$

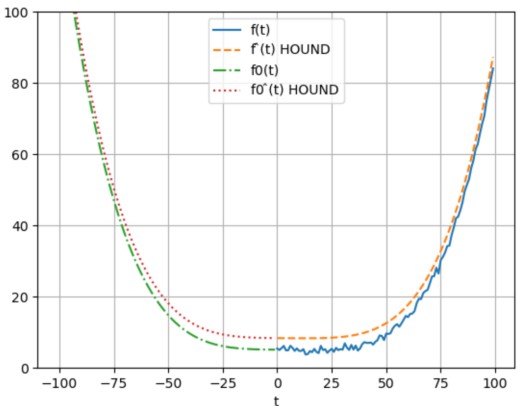

Figure 2: The differentiable signal $f(t)$ and its interpolation $\hat{f}(t)$, extrapolation of the signal $f_0(t)$ and its estimates $\hat{f}_0(t)$ beyond the beginning of the range

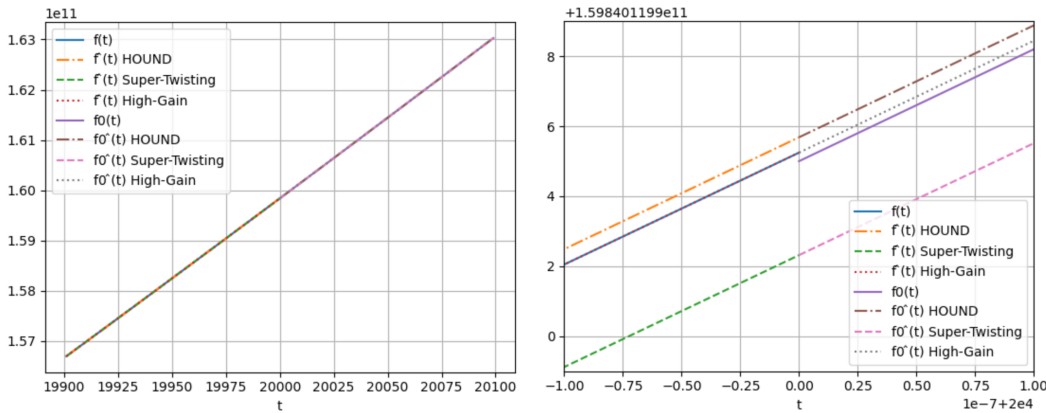

Figure 3: The differentiable signal $f(t)$ and its interpolation $\hat{f}(t)$, extrapolation of the signal $f_0(t)$ and its estimates $\hat{f}_0(t)$ beyond the end of the range

By choosing $\Delta t = 1$, the algorithm simplifies to the following form:

$$
\begin{cases}
z_0[t] = \frac{25}{t}\varepsilon[t] + z_0[t-1] + z_1[t-1] + \frac{z_2[t-1]}{2} + \frac{z_3[t-1]}{6} + \frac{z_4[t-1]}{24} \\
z_1[t] = \frac{300}{t^2}\varepsilon[t] + z_1[t-1] + z_2[t-1] + \frac{z_3[t-1]}{2} + \frac{z_4[t-1]}{6} \\
z_2[t] = \frac{2100}{t^3}\varepsilon[t] + z_2[t-1] + z_3[t-1] + \frac{z_4[t-1]}{2} \\
z_3[t] = \frac{8400}{t^4}\varepsilon[t] + z_3[t-1] + z_4[t-1] \\
z_4[t] = \frac{15120}{t^5}\varepsilon[t] + z_4[t-1] \\
\varepsilon[t] = f[t] - \left(z_0[t-1] + z_1[t-1] + \frac{z_2[t-1]}{2} + \frac{z_3[t-1]}{6} + \frac{z_4[t-1]}{24}\right)
\end{cases}
,
$$

where $z_0[0] = f[0]$, $z_1[0] = 0$, $z_2[0] = 0$, $z_3[0] = 0$, $z_4[0] = 0$ and $t = 1, 2, 3, \ldots, 20000$.

For comparison, we present a analogous high-gain differentiation algorithm, also for $\Delta t = 1$:

$$
\begin{cases}
x_0[t] = \frac{5}{5^1}\varepsilon[t] + x_0[t-1] + x_1[t-1] + \frac{x_2[t-1]}{2} + \frac{x_3[t-1]}{6} + \frac{x_4[t-1]}{24} \\
x_1[t] = \frac{10.03}{5^2}\varepsilon[t] + x_1[t-1] + x_2[t-1] + \frac{x_3[t-1]}{2} + \frac{x_4[t-1]}{6} \\
x_2[t] = \frac{9.30}{5^3}\varepsilon[t] + x_2[t-1] + x_3[t-1] + \frac{x_4[t-1]}{2} \\
x_3[t] = \frac{4.57}{5^4}\varepsilon[t] + x_3[t-1] + x_4[t-1] \\
x_4[t] = \frac{1.1}{5^5}\varepsilon[t] + x_4[t-1] \\
\varepsilon[t] = f[t] - \left(x_0[t-1] + x_1[t-1] + \frac{x_2[t-1]}{2} + \frac{x_3[t-1]}{6} + \frac{x_4[t-1]}{24}\right)
\end{cases}
,
$$

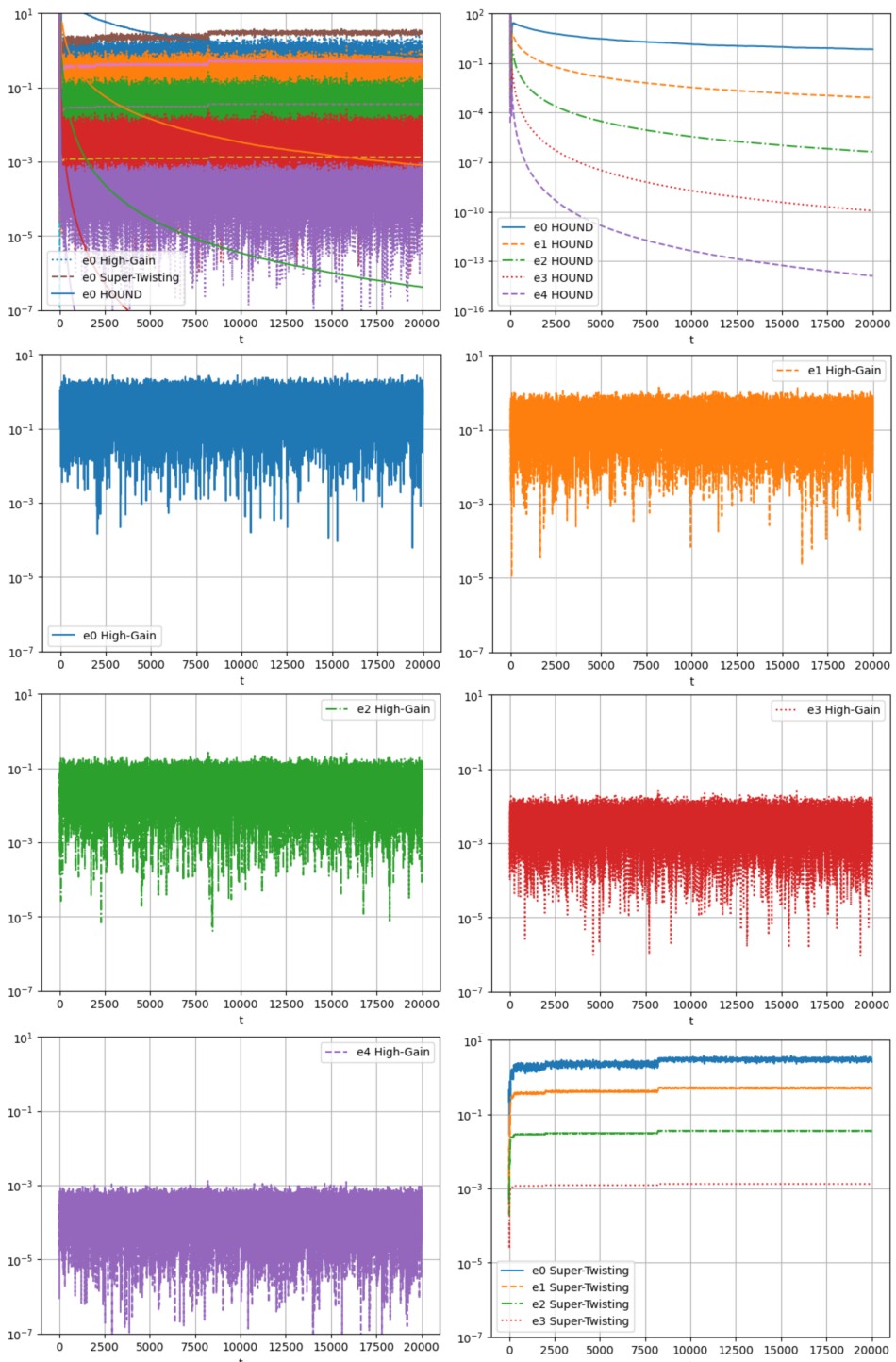

Figure 4: Errors of estimation of the signal and its derivatives in logarithmic scale (absolute values)

under those initial conditions $x_0[0] = f[0]$, $x_1[0] = 0$, $x_2[0] = 0$, $x_3[0] = 0$, $x_4[0] = 0$.

For comparative analysis, we also present the super-twisting differentiation algorithm, also with a sampling interval $\Delta t = 1$. Since its optimal tuning requires knowledge of the Lipschitz constant, and for the signal under consideration $f_0^{(5)}(t) \equiv 0$, we employ a differentiator of one order lower.

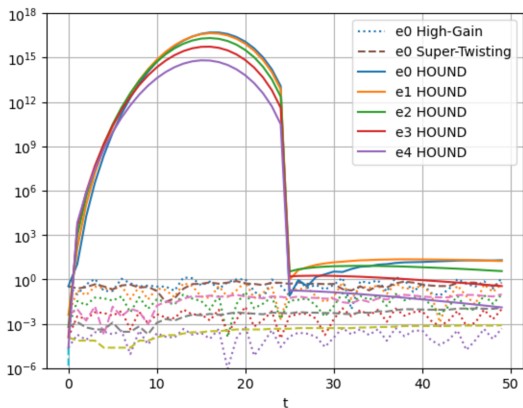

Figure 5: Errors of estimation of signal and its derivatives in logarithmic scale (absolute values), initial range

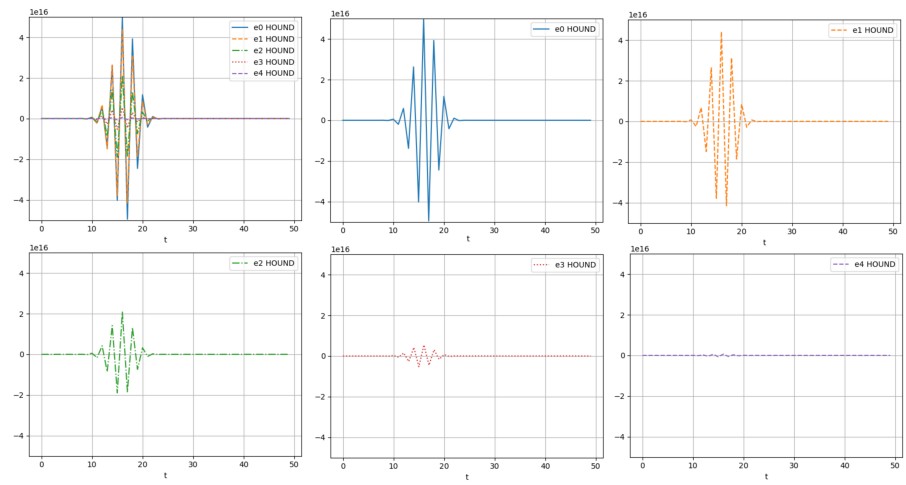

Figure 6: Estimation errors of the signal and its derivatives, initial range

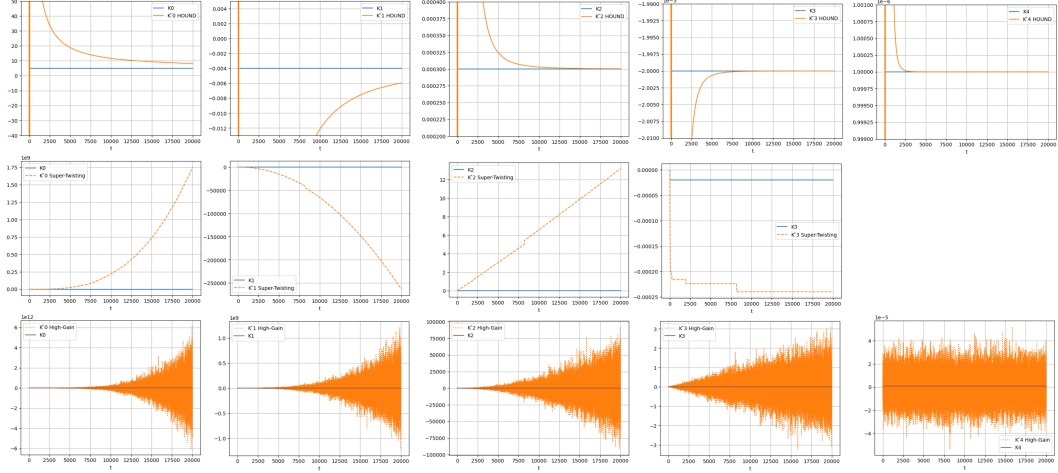

Figure 7: Convergence of "learned parameters" of regression to the coefficients of the polynomial

Accordingly, we set $L = |\frac{f_0^{(4)}(t)}{1.1}| = |\frac{4!K_4}{1.1}| = 24\frac{0.000001}{1.1}$ and define $y_4[t] \equiv 4!K_4 = 0.000024$ with the coefficients taken from Table 2 in Mojallizadeh et al. (2021):

$$\begin{cases} y_0[t] = \quad 3L^{\frac{1}{4}}|\varepsilon[t]|^{\frac{3}{4}}sign(\varepsilon[t]) + y_0[t-1] + y_1[t-1] + \frac{y_2[t-1]}{2} + \frac{y_3[t-1]}{6} \\ y_1[t] = 4.16L^{\frac{2}{4}}|\varepsilon[t]|^{\frac{2}{4}}sign(\varepsilon[t]) + y_1[t-1] + y_2[t-1] + \frac{y_3[t-1]}{2} \\ y_2[t] = 3.06L^{\frac{3}{4}}|\varepsilon[t]|^{\frac{1}{4}}sign(\varepsilon[t]) + y_2[t-1] + y_3[t-1] \\ y_3[t] = \quad 1.1L^{\frac{4}{4}}|\varepsilon[t]|^{\frac{0}{4}}sign(\varepsilon[t]) + y_3[t-1] \\ \varepsilon[t] = f[t] - (y_0[t-1] + y_1[t-1] + \frac{y_2[t-1]}{2} + \frac{y_3[t-1]}{6}) \end{cases},$$

under similar initial conditions $y_0[0] = f[0]$, $y_1[0] = 0$, $y_2[0] = 0$, $y_3[0] = 0$.

After completing the calculations at $t = 20000$, the results for HOUND differentiator are $z_0[20000] = 159840119925.682$, $z_1[20000] = 31976011.9968$, $z_2[20000] = 4797.601$, $z_3[20000] = 0.47988$, $z_4[20000] = 0.000024$. For high-gain differentiator the results are $x_0[20000] = 159840119925.244$, $x_1[20000] = 31976011.9772$, $x_2[20000] = 4797.597$, $x_3[20000] = 0.47983$, $x_4[20000] = 0.000045$. For super-twisting method the results are $y_0[20000] = 159840119922.316$, $y_1[20000] = 31976011.5038$, $y_2[20000] = 4797.565$, $y_3[20000] = 0.47856$, $y_4[20000] \equiv 0.000024$.

We substitute these values into the approximation formula (Equation (6)) and vary $\tau$ in the range from -100 to 20100. The interval from 0 to 20000 corresponds to interpolation (Figure (1)), while the ranges from $-100$ to 0 (Figure (2)) and from 20000 to 20100 (Figure (3)) show extrapolation.

During the calculations, we compute the errors in the estimates of the signal and its derivatives as follows: $e_0[t] = z_0[t] - f_0[t]$, $e_1[t] = z_1[t] - f_0^{(1)}[t]$, $e_2[t] = z_2[t] - f_0^{(2)}[t]$, $e_3[t] = z_3[t] - f_0^{(3)}[t]$, $e_4[t] = z_4[t] - f_0^{(4)}[t]$ (instead of $z$ for high-gain we substitute $x$, and for super-twisting we substitute $y$). Figure (4) shows the decrease in the absolute values of these errors as $t$ increases, while Figure (5) displays the transient phase. Figure (6) illustrates the signed error values over time.

Substituting the calculation results into formula (3) allows us to estimate the constants that define the polynomial expression of the unnoised signal $f_0(t)$. The estimated values are: $\hat{K}_0 = 8.25$, $\hat{K}_1 = -0.005977$, $\hat{K}_2 = 0.0003$, $\hat{K}_3 = -0.00002$, $\hat{K}_4 = 0.000001$. Figure (7) shows the process of convergence to these true values. For both the high-gain differentiator and super-twisting differentiator, the estimates of the polynomial approximation coefficients fail to converge to constant values.

The results demonstrate strong interpolation accuracy across the entire range of data processed by the HOUND differentiator, highlighting the effectiveness of cumulative smoothing. Additionally, the accuracy of the approximation does not degrade near the boundaries of the $t$-range, enabling reliable extrapolation. For the high-gain differentiator and super-twisting differentiator, the interpolation accuracy deteriorates rapidly in the initial segment of data range. Consequently extrapolation is feasible only in one direction, namely beyond the end of the range (i.e. into the future for time-dependent function).

The presented cumulative smoothing algorithm also performs well in estimating the polynomial signal and its derivatives. Interestingly, the higher the order of the derivative, the more accurate the estimates become, as the error decreases more rapidly for higher-order derivatives. Additionally, the algorithm effectively filters out additive white Gaussian noise, leading to accurate estimates of the polynomial representation of the unnoised signal. For the other two algorithms, the error remains low but does not decrease throughout the computations, persisting at an almost constant level that precludes the possibility of performing a polynomial approximation. Beyond a certain iteration, the error produced by the proposed algorithm (HOUND) becomes lower than that of the compared algorithms (the high-gain differentiator and the super-twisting differentiator), which are only able to filter noise to a limited extent.

For the HOUND differentiator the estimation process for both the signal and its derivatives includes a distinctly nonlinear phase, characterized by a sharp rise in values, followed by an equally rapid decrease. This results in the estimates stabilizing near their true values and the errors approaching zero. During this transient phase, the sign of the errors changes with each step of the calculation. Once the transient phase concludes, the errors begin to reduce smoothly, with a sublinear rate of convergence.

Another demostration cases are given in the Appendices B and C. The software code to reproduce the demos is available via the repository `https://github.com/katrichek/HOUND/`

## 4 CONCLUSION

The universal numerical differentiator introduced here efficiently estimates the derivatives of a function (or signal, when the function is time-dependent), even in the presence of additive noise. It performs polynomial approximations of a selected order. The proposed cumulative smoothing algorithm operates in a recurrent form, allowing for real-time data processing. This method requires no trainable parameters and involves only one hyperparameter: the integer order of the differentiator. Overestimating this order doesn't negatively affect performance, aside from increasing computational load. The method doesn't require knowledge of the Lipschitz constant for the highest estimated derivative, and it offers theoretical guarantees of convergence (in the continuous case) if boundedness is ensured.

Future research on this algorithm could explore several key areas. First, it would be beneficial to compare different discretization methods for continuous system of nonlinear differential equations (Equation (1)). Second, proving the stability of the discrete form of the universal numerical differentiator (Equation (7)) and eliminating the transient phase would be valuable. The impact of noise variance in the signal and the irregularity of time steps and initial conditions on the accuracy and speed of differentiation also merits investigation. Other areas of interest include mitigating the effects of noise in the derivatives of the signal, addressing the accumulation of computational errors (particularly when dealing with large numbers or factorial operations), and exploring parallelization of the differentiation process for large datasets, possibly through matrix transformations to enable hardware-accelerated calculations. Finally, developing cumulative smoothing techniques for multivariate high-dimensional data and signals with periodic components, is another promising direction.

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

## A   APPENDIX

Let us now solve the system of differential equations (Equation (1)). First, we express the estimation errors for the signal and its derivatives as:

$$e_{m-1}(t) = z_{m-1}(t) - f^{(m-1)}(t). \tag{10}$$

Substituting into system (1), the equations take the form:

$$e_{m-1}^{(1)}(t) = e_m(t) - \frac{(n+m-1)!}{m!(n-m)!}\frac{n}{t^m}e_0(t). \tag{11}$$

For $n = 1, 2, 3$, here are examples of system (11) (omitting the explicit dependence on $t$):

$$e_0^{(1)} = -\frac{1}{t}e_0 - f^{(1)} \qquad \text{if } n = 1,$$

$$\begin{cases} e_0^{(1)} = -\frac{4}{t}e_0 + e_1 \\ e_1^{(1)} = -\frac{6}{t^2}e_0 - f^{(2)} \end{cases} \qquad \text{if } n = 2,$$

$$\begin{cases} e_0^{(1)} = -\frac{9}{t}e_0 + e_1 \\ e_1^{(1)} = -\frac{36}{t^2}e_0 + e_2 \\ e_2^{(1)} = -\frac{60}{t^3}e_0 - f^{(3)} \end{cases} \qquad \text{if } n = 3.$$

To solve each equation in system (11), we differentiate them $d$ times, where $d = n - m$. Using Leibniz's rule for differentiating a product, we have:

$$\left(\frac{1}{t^m}e_0(t)\right)^{(d)} = \sum_{k=0}^{d}\binom{d}{k}\left(\frac{1}{t^m}\right)^{(d-k)}e_0^{(k)}(t) = \sum_{k=0}^{d}\binom{d}{k}(-1)^{d-k}\frac{(m+d-k-1)!}{(m-1)!}\frac{1}{t^{m+d-k}}e_0^{(k)}(t).$$

After differentiating, the system (11) becomes:

$$e_{m-1}^{(n-m+1)}(t) = e_m^{(n-m)}(t) - \frac{(n+m-1)!n}{m!(n-m)!}\sum_{k=0}^{n-m}(-1)^{n-m-k}\binom{n-m}{k}\frac{(n-1-k)!}{(m-1)!}\frac{1}{t^{n-k}}e_0^{(k)}(t).$$

Let us write this expression as:

$$e_{m-1}^{(n-m+1)}(t) = e_m^{(n-m)}(t) + X_{n,m},$$

where $e_0^{(n)}(t) = X_{n,1} + e_1^{(n-1)}(t)$, $e_1^{(n-1)}(t) = X_{n,2} + e_2^{(n-2)}(t), \ldots, e_{n-2}^{(2)}(t) = X_{n,n-1} + e_{n-1}^{(1)}(t)$, $e_{n-1}^{(1)}(t) = X_{n,n} + e_n(t)$. After performing the stroboscopic summation, we obtain $e_0^{(n)}(t) = e_n(t) + \sum_{m=1}^{n}X_{n,m}$.

This simplifies to:

$$e_0^{(n)}(t) = e_n(t) - \sum_{m=1}^{n}\frac{(n+m-1)!n}{m!(n-m)!}\sum_{k=0}^{n-m}(-1)^{n-m-k}\frac{(n-m)!}{k!(n-m-k)!}\frac{(n-1-k)!}{(m-1)!}\frac{1}{t^{n-k}}e_0^{(k)}(t).$$

Next, we group the summands by the order of the derivative:

$$e_0^{(n)}(t) = e_n(t) - \sum_{d=0}^{n-1}\frac{1}{t^{n-d}}e_0^{(d)}(t)\sum_{m=1}^{n-d}(-1)^{n-m-d}\frac{(n+m-1)!n}{m!(n-m)!}\frac{(n-m)!}{d!(n-m-d)!}\frac{(n-1-d)!}{(m-1)!}.$$

We sum $m$ from 0, keeping in mind that $\frac{1}{(-1)!} = 0$. This gives:

$$e_0^{(n)}(t) = e_n(t) - \sum_{d=0}^{n-1}\frac{n!}{d!}\frac{1}{t^{n-d}}e_0^{(d)}(t)\sum_{m=0}^{n-d}(-1)^{n-m-d}\binom{n+m-1}{m}\binom{n-1-d}{n-m-d}.$$

Using the identity $\binom{n+m-1}{m} = (-1)^m \binom{-n}{m}$, Vandermonde's identity $\sum_{m=0}^{n-d} \binom{-n}{m} \binom{n-1-d}{n-d-m} = \binom{-1-d}{n-d} n - d$, and identities $\binom{-1-d}{n-d} = (-1)^{n-d} \binom{n}{n-d}$, $\binom{n}{n-d} = \binom{n}{d}$ taken from Graham et al. (1994), the sum simplifies to:

$$e_0^{(n)}(t) = e_n(t) - \sum_{d=0}^{n-1} \frac{n!}{d!} \binom{n}{d} \frac{1}{t^{n-d}} e_0^{(d)}(t).$$

Multiplying the equation by $t^n$ and using the fact that $e_n(t) = -f^{(n)}(t)$, according to (10) (since, $z_n(t) \equiv 0$), we get:

$$\sum_{d=0}^{n} \frac{n!}{d!} \binom{n}{d} t^d e_0^{(d)}(t) = -t^n f^{(n)}(t). \tag{12}$$

This is an Euler differential equation with the characteristic polynomial:

$$\sum_{d=0}^{n} \frac{n!}{d!} \binom{n}{d} \prod_{k=0}^{d-1} (\lambda - k) = 0. \tag{13}$$

Here are examples of Equation (12) for $n = 1, 2, 3$ (we omit the explicit dependence on $t$ for brevity):

$$t e_0^{(1)} + e_0 = -t f^{(1)}(t) \quad \text{if } n = 1,$$

$$t^2 e_0^{(2)} + 4 t e_0^{(1)} + 2 e_0 = -t^2 f^{(2)}(t) \quad \text{if } n = 2,$$

$$t^3 e_0^{(3)} + 9 t^2 e_0^{(2)} + 18 t e_0^{(1)} + 6 e_0 = -t^3 f^{(3)}(t) \quad \text{if } n = 3.$$

The corresponding characteristic polynomials (13) are as follows:

$$\lambda + 1 = 0 \quad \text{if } n = 1,$$

$$\lambda(\lambda - 1) + 4\lambda + 2 = \lambda^2 + 3\lambda + 2 = (\lambda + 1)(\lambda + 2) = 0 \quad \text{if } n = 2,$$

$$\lambda(\lambda-1)(\lambda-2) + 9\lambda(\lambda-1) + 18\lambda + 6 = \lambda^3 + 6\lambda^2 + 11\lambda + 6 = (\lambda+1)(\lambda+2)(\lambda+3) = 0 \quad \text{if } n = 3.$$

Let us express equation (13) using the Stirling numbers of the first kind (signed) $\prod_{k=0}^{d-1}(\lambda - k) = \sum_{k=0}^{d} s(d, k) \lambda^k$:

$$\sum_{d=0}^{n} \frac{n!}{d!} \binom{n}{d} \sum_{k=0}^{d} s(d, k) \lambda^k = 0.$$

We now group the summands by powers of $\lambda$ and extend the summation limits from 0 to $n$, since $s(d, k) = 0$ for $k > d$:

$$\sum_{k=0}^{n} \lambda^k \sum_{d=0}^{n} \frac{n!}{d!} \binom{n}{d} s(d, k) = 0.$$

According to identity 21 of Boyadzhiev (2021) and the relationship between the Stirling numbers of the first kind with and without sign, the inner sum can be simplified as:

$$\sum_{d=0}^{n} \frac{n!}{d!} \binom{n}{d} s(d, k) = (-1)^{n-k} s(n+1, k+1) = \begin{bmatrix} n+1 \\ k+1 \end{bmatrix}.$$

Then the characteristic polynomial (13):

$$\sum_{k=0}^{n} \begin{bmatrix} n+1 \\ k+1 \end{bmatrix} \lambda^k = 0$$

is written as follows:

$$\prod_{k=1}^{n} (\lambda + k) = 0. \tag{14}$$

Then the general solution of the equation (12) according to 5.1.2.39 from Polyanin & Zaitsev (2003):

$$e_0(t) = \sum_{d=1}^{n} \frac{C_d}{t^d}. \tag{15}$$

Substituting (15) into the system (11), it can be shown that its general solution has the form:

$$e_{m-1}(t) = \sum_{d=1}^{n} \frac{a_{m,d,n}}{t^{d+m-1}} C_d, \tag{16}$$

where the coefficients $a_{m,d,n} > 0$ are calculated via the recurrence relation:

$$a_{m+1,d,n} = -(d+m-1)a_{m,d,n} + \frac{(n+m-1)!n}{m!(n-m)!}, \tag{17}$$

where $a_{1,d,n} = 1$ for $m = 1$, and $a_{k,d,n} = 0$ when $k > n$.

To obtain the particular solution, we vary $C_d(t)$.

Substituting the general solution (11) and its derivative into the system (16):

$$e_{m-1}^{(1)}(t) = \sum_{d=1}^{n} a_{m,d,n} \left( \frac{C_d^{(1)}(t)}{t^{d+m-1}} - (d+m-1)\frac{C_d(t)}{t^{d+m}} \right),$$

it can be shown that the summands with $C_d(t)$ vanish, and we are left with a system of equations:

$$\sum_{d=1}^{n} a_{m,d,n} \frac{C_d^{(1)}(t)}{t^{d+m-1}} = \begin{cases} 0, m < n \\ -f^{(n)}(t), m = n \end{cases}.$$

Multiplying it by $t^{n+m-1}$, we obtain:

$$\sum_{d=1}^{n} a_{m,d,n} t^{n-d} C_d^{(1)}(t) = \begin{cases} 0, m < n \\ -t^{2n-1} f^{(n)}(t), m = n \end{cases},$$

which can be written in matrix form:

$$\mathbf{A}_{n \times n} \mathbf{D}_{n \times n} \mathbf{C}_{n \times 1} = \mathbf{E}_{n \times 1}, \tag{18}$$

where the matrix $\mathbf{A}_{m,d}$ consists of elements $a_{m,d,n}$, the matrix $\mathbf{D}_{d,d}$ contains only the main diagonal with expressions $t^{n-d}$ (the other elements are zero), vector $\mathbf{C}_d$ consists of elements $C_d^{(1)}(t)$, and vector $\mathbf{E}$ is zero except for the last element $\mathbf{E}_n = -t^{2n-1} f^{(n)}(t)$. In detailed form, the system (18) looks like this:

$$\begin{bmatrix} a_{1,1,n} & a_{1,2,n} & \cdots & a_{1,n-1,n} & a_{1,n,n} \\ a_{2,1,2} & a_{2,2,n} & \cdots & a_{2,n-1,n} & a_{2,n,n} \\ \vdots & \vdots & \ddots & \vdots & \vdots \\ a_{n-1,1,n} & a_{n-1,2,n} & \cdots & a_{n-1,n-1,n} & a_{n-1,n,n} \\ a_{n,1,n} & a_{n,2,n} & \cdots & a_{n,n-1,n} & a_{n,n,n} \end{bmatrix} \begin{bmatrix} t^{n-1} & 0 & \cdots & 0 & 0 \\ 0 & t^{n-2} & \cdots & 0 & 0 \\ \vdots & \vdots & \ddots & \vdots & \vdots \\ 0 & 0 & \cdots & t & 0 \\ 0 & 0 & \cdots & 0 & 1 \end{bmatrix} \begin{bmatrix} C_1^{(1)}(t) \\ C_2^{(1)}(t) \\ \cdots \\ C_{n-1}^{(1)}(t) \\ C_n^{(1)}(t) \end{bmatrix} = \begin{bmatrix} 0 \\ 0 \\ \cdots \\ 0 \\ -t^{2n-1} f^{(n)}(t) \end{bmatrix}.$$

Let us give examples of (18) for $n = 1, 2, 3$ (we omit for brevity the explicit statement of the dependence on $t$):

$$[1][1]\left[C_1^{(1)}\right] = \left[-tf^{(1)}\right] \quad \text{if } n = 1,$$

$$\begin{bmatrix} 1 & 1 \\ 3 & 2 \end{bmatrix} \begin{bmatrix} t & 0 \\ 0 & 1 \end{bmatrix} \begin{bmatrix} C_1^{(1)} \\ C_2^{(1)} \end{bmatrix} = \begin{bmatrix} 0 \\ -t^3 f^{(2)} \end{bmatrix} \quad \text{if } n = 2,$$

$$\begin{bmatrix} 1 & 1 & 1 \\ 8 & 7 & 6 \\ 20 & 15 & 12 \end{bmatrix} \begin{bmatrix} t^2 & 0 & 0 \\ 0 & t & 0 \\ 0 & 0 & 1 \end{bmatrix} \begin{bmatrix} C_1^{(1)} \\ C_2^{(1)} \\ C_3^{(1)} \end{bmatrix} = \begin{bmatrix} 0 \\ 0 \\ -t^5 f^{(3)} \end{bmatrix} \quad \text{if } n = 3.$$

It can be shown that the determinant of the product of matrices $\mathbf{A}_{n \times n} \mathbf{D}_{n \times n}$ is expressed through the superfactorial:

$$|\mathbf{AD}| = (-1)^{\lfloor \frac{n}{2} \rfloor} t^{\frac{n(n-1)}{2}} \prod_{k=0}^{n-1} k!,$$

hence the system of equations (18) is nonsingular. It can be shown that its solution has the form:

$$C_d^{(1)}(t) = \frac{(-1)^d}{b_{d,n}} t^{d+n-1} f^{(n)}(t), \tag{19}$$

where the coefficients $b_{d,n} > 0$ are calculated by the formula:

$$b_{d,n} = (n-d)!(d-1)!, \tag{20}$$

then by integrating (19) and by substituting into (16), we obtain the particular solution of the system (11):

$$e_{m-1}(t) = \sum_{d=1}^{n} \frac{a_{m,d,n}}{t^{d+m-1}} \left( c_d + \frac{(-1)^d}{b_{d,n}} \int_{t_0}^{t} \tau^{d+n-1} f^{(n)}(\tau) \, d\tau \right). \tag{21}$$

By expanding the errors $e_{m-1}(t)$ in terms of the difference between the estimated derivatives and the actual derivatives of the signal (10), we derive the final expression in Equation (2).

Let's now apply integration by parts to the integral in Equation (21):

$$\int_{t_0}^{t} \tau^{d+n-1} f^{(n)}(\tau) \, d\tau = \tau^{d+n-1} f^{(n-1)}(\tau) \Big|_{t_0}^{t} - \int_{t_0}^{t} (\tau^{d+n-1})^{(1)} f^{(n-1)}(\tau) \, d\tau =$$

$$t^{d+n-1} f^{(n-1)}(t) - t_0^{d+n-1} f^{(n-1)}(t_0) - (d+n-1) \int_{t_0}^{t} \tau^{d+n-2} f^{(n-1)}(\tau) \, d\tau.$$

Next, we substitute the variable $u = \tau^{d+n-1}$ in the remaining integral:

$$(d+n-1) \int_{t_0}^{t} \tau^{d+n-2} f^{(n-1)}(\tau) \, d\tau = \int_{t_0^{d+n-1}}^{t^{d+n-1}} f^{(n-1)} \left( u^{\frac{1}{d+n-1}} \right) du.$$

If the following condition holds:

$$|f^{(n-1)}(x)| \leq L, \tag{22}$$

where $L$ is a non-negative Lipschitz constant, then integral is bounded by:

$$\left| \int_{t_0^{d+n-1}}^{t^{d+n-1}} f^{(n-1)} \left( u^{\frac{1}{d+n-1}} \right) du \right| \leq t^{d+n-1} L + t_0^{d+n-1} L.$$

Therefore, the integral in Equation (21) is bounded by:

$$\left| \int_{t_0}^{t} \tau^{d+n-1} f^{(n)}(\tau) \, d\tau \right| \leq t^{d+n-1} 2L + t_0^{d+n-1} 2L.$$

Consequently, the estimation error (21) is bounded by:

$$|e_{m-1}(t)| \leq t^{n-m} 2L \left| \sum_{d=1}^{n} \frac{(-1)^d}{b_{d,n}} a_{m,d,n} \right| + \sum_{d=1}^{n} \frac{a_{m,d,n}}{t^{d+m-1}} \left( |c_d| + \frac{t_0^{d+n-1} 2L}{b_{d,n}} \right).$$

It can be shown that the first sum in the previous expression equals zero for all $m$ except when $m = n$:

$$\sum_{d=1}^{n} \frac{(-1)^d}{b_{d,n}} a_{m,d,n} = \begin{cases} 0 & \text{if } m < n \\ -1 & \text{if } m = n \end{cases}.$$

Thus, the error in estimating the signal and its derivatives (21) is bounded by:

$$|e_{m-1}(t)| \leq 2L + \sum_{d=1}^{n} \frac{a_{m,d,n}}{t^{d+m-1}} \left( |c_d| + \frac{t_0^{d+n-1} 2L}{b_{d,n}} \right), \tag{23}$$

which tends to the constant $2L$, as $t$ increases.

The correctness of the formulas (17) and (20), which calculate the constants $a_{m,d,n}$, $b_{d,n}$ and all nd the subsequent conclusions for any values of $n$, requires further formal proof. However, in this paper, we have verified the accuracy of all the given formulas across a wide range of values for $n$.

# B   HARMONIC CASE

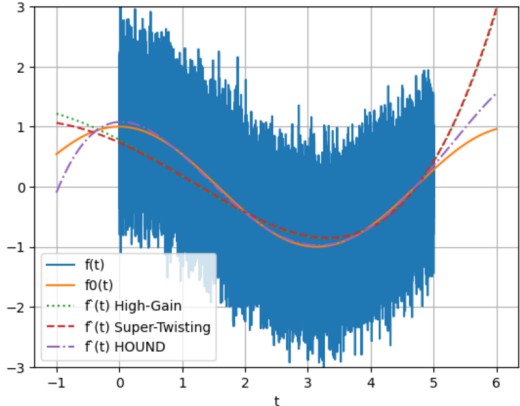

Figure 8: The harmonic signal $f(t)$ and its interpolation and extrapolation

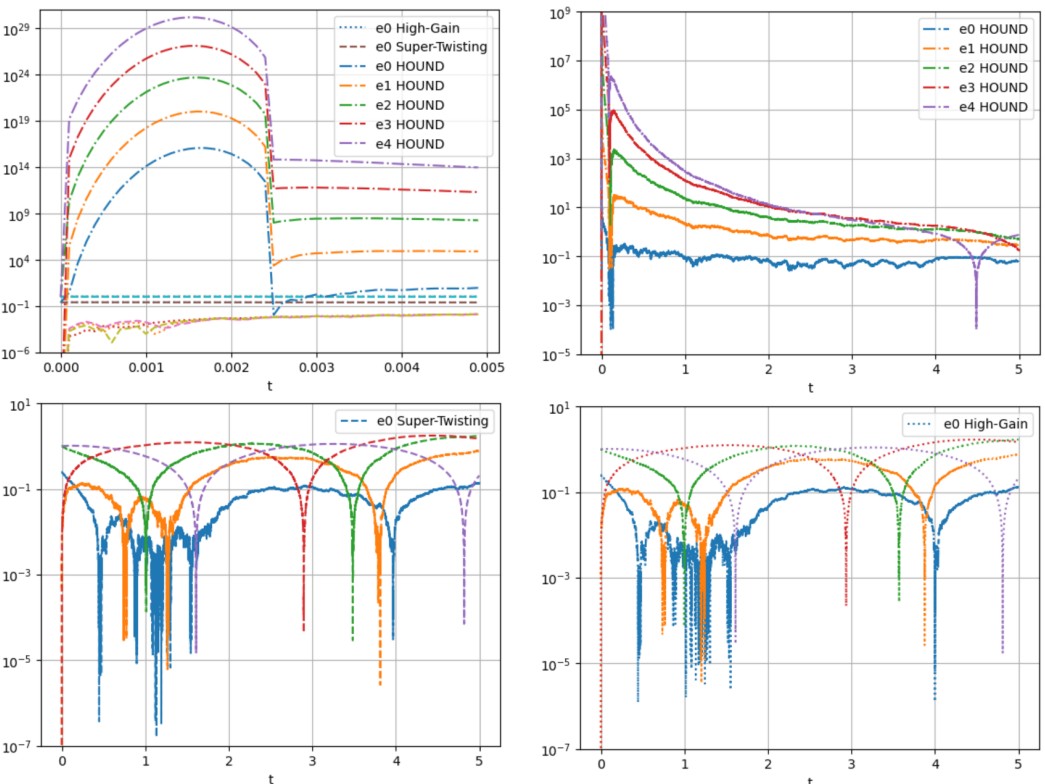

Figure 9: Errors of estimation of the signal and its derivatives in logarithmic scale (absolute values)

Consider a harmonic signal with high-frequency disturbance $f(t) = f_0(t) + \eta_0(t) = cos(t) + (sin(10000t) + N(0,1))/2$, in the range $t =$ from 0 to 5. We apply differentiators with $n = 5$ and $\Delta t = 0.0001$. For the high-gain differentiator and super-twisting differentiator, we choose the Lipschitz constant $L = 1$. The results are shown in Figures (8) and (9).

# C  NONSMOOTH CASE

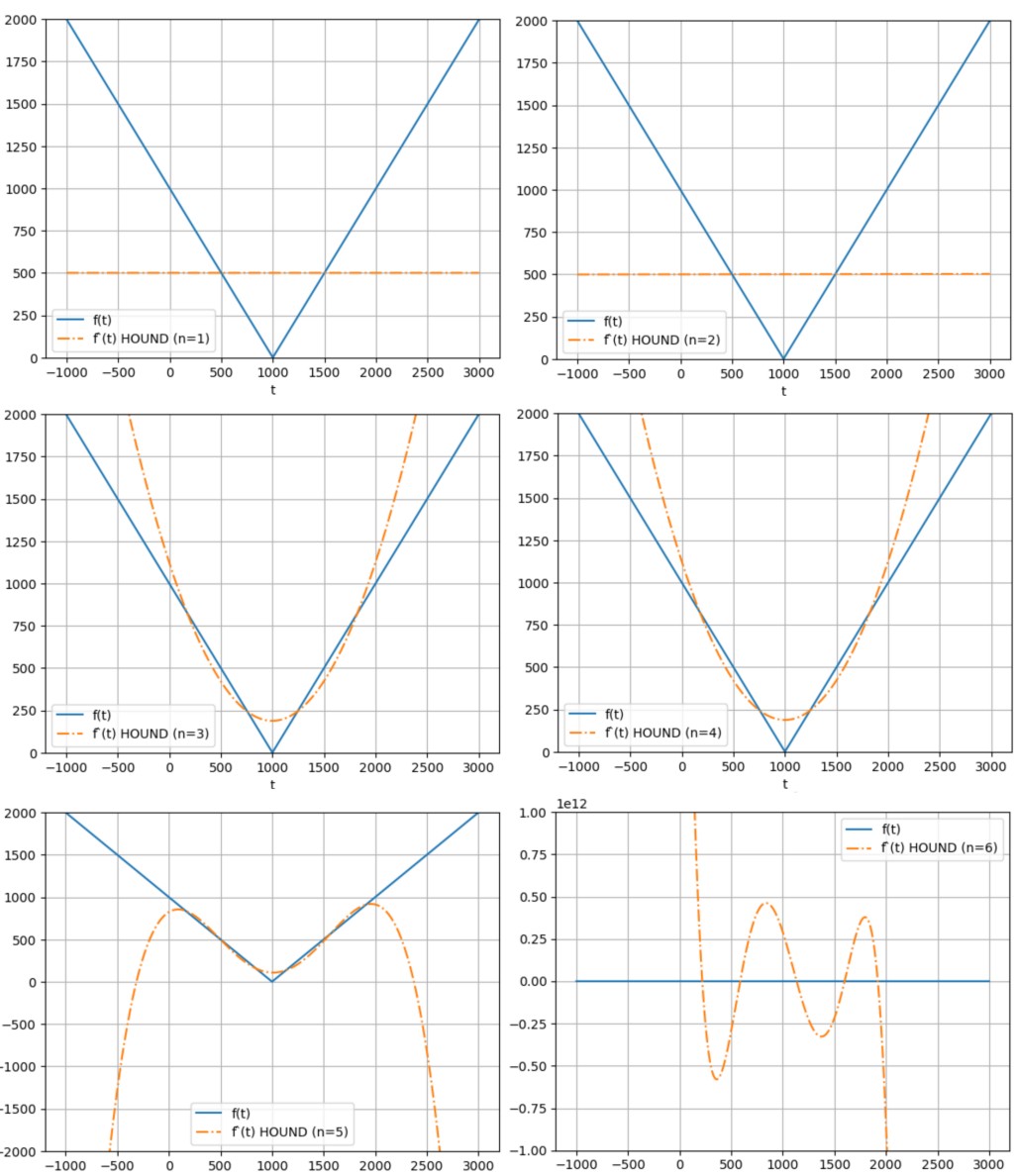

Figure 10: The nonsmooth signal $f(t)$ and its interpolation and extrapolation

Consider a nonsmooth signal $f(t) = abs(t - 1000)$ in the range $t =$ from 0 to 2000. We apply HOUND differentiator with $n$ from 1 to 6, and $\Delta t = 1$. The results are shown at Figure (10).

