# OpenReview forum: "HOUND: High-Order Universal Numerical Differentiator for a Parameter-free Polynomial Online Approximation"
_mathai.club/MathAI/2025/Conference — MathAI 2025 Oral_

### Official Review · Reviewer_zH7T · 2025-02-25
**The article introduces HOUND (High-Order Universal Numerical Differentiator), a novel, parameter-free method for estimating high-order derivatives of signals, even in the presence of additive noise. HOUND stands out for its versatility, requiring no prior knowledge of signal characteristics or noise properties, and is applicable to real-time signal processing and control systems. The authors provide rigorous mathematical derivations, a practical discretization method, and a comprehensive comparison with existing techniques, demonstrating HOUND's superior performance in interpolation, extrapolation, and noise filtering. While the method shows great promise, the article could benefit from further analysis of computational complexity, empirical validation on real-world data, and exploration of its stability and transient behavior. Overall, HOUND represents a significant advancement in numerical differentiation, with broad applicability across various fields.**

**Rating:** 9
**Confidence:** 2

**Review:**

General Overview:
The article presents a novel numerical differentiator, termed HOUND (High-Order Universal Numerical Differentiator), designed for high-order derivative estimation of signals, particularly in the presence of additive white/Gaussian noise. The authors propose a system of nonlinear differential equations that can estimate derivatives up to any arbitrary order without requiring prior knowledge of the signal's Lipschitz constant or noise characteristics. The method is parameter-free, making it highly versatile for various applications in signal processing and control systems. The paper also introduces a discretization method for implementing the differentiator in real-time, which is capable of handling both interpolation and extrapolation tasks without the need for data accumulation or coefficient fitting.

Strengths of the Article:

1. Innovative Approach:
   The proposed HOUND differentiator is a significant advancement in the field of numerical differentiation. Unlike traditional methods that require tuning based on signal characteristics, HOUND operates without any parameter adjustments, making it a universal tool for a wide range of signals. This is particularly useful in real-time applications where signal characteristics may not be known a priori.

2. Mathematical Rigor:
   The article is mathematically rigorous, providing detailed derivations of the system of differential equations and their solutions. The authors have thoroughly analyzed the convergence properties of the differentiator, demonstrating that the error converges to zero for polynomial signals with additive noise and remains bounded for more general signals. The theoretical guarantees provided in the paper are robust and well-supported by mathematical proofs.

3. Practical Implementation:
   The discretization method proposed for the HOUND differentiator is practical and efficient. The authors have developed a cumulative smoothing algorithm that operates online, making it suitable for real-time applications. The algorithm's ability to handle irregular time steps and its performance in filtering out additive white Gaussian noise are particularly noteworthy.

4. Comparative Analysis:
   The article includes a comprehensive comparison with existing methods such as the super-twisting and high-gain differentiators. The results demonstrate that HOUND outperforms these methods in terms of interpolation accuracy, extrapolation capability, and noise filtering. The comparative analysis is thorough and provides strong evidence of the superiority of the proposed method.

5. Broad Applicability:
   The HOUND differentiator is not limited to specific types of signals. It can handle polynomial signals, harmonic signals, and signals corrupted by noise. The method's ability to automatically perform polynomial regression on the differentiable signal further extends its applicability in machine learning and data analysis.

Weaknesses and Areas for Improvement:

1. Computational Complexity:
   While the article discusses the theoretical aspects of the HOUND differentiator in detail, it does not provide a thorough analysis of the computational complexity of the proposed method. Given that the differentiator involves high-order derivatives and factorial operations, it is essential to evaluate its computational efficiency, especially for large datasets or real-time applications with high sampling rates.

2. Empirical Validation:
   Although the article presents theoretical results and comparative analysis, it lacks extensive empirical validation on real-world datasets. While the authors provide examples with synthetic signals, it would be beneficial to see the performance of HOUND on real-world signals, such as those from engineering, finance, or biological systems. This would further validate the method's practical utility.

3. Noise in Derivatives:
   The article primarily focuses on filtering additive noise in the signal itself. However, in many practical scenarios, the derivatives of the signal may also be corrupted by noise. The authors briefly mention this issue but do not provide a detailed analysis or solution. Extending the HOUND differentiator to handle noise in the derivatives would enhance its applicability.

4. Initial Conditions and Transient Behavior:
   The article assumes initial conditions for the differentiator, but it does not thoroughly discuss the impact of these initial conditions on the transient behavior of the system. In practical applications, the choice of initial conditions can significantly affect the performance of the differentiator, especially in the initial phase of operation. A more detailed analysis of this aspect would be beneficial.

5. Generalization to Higher Dimensions:
   The article focuses on scalar signals, and the proposed differentiator is designed for one-dimensional data. However, many real-world applications involve multi-dimensional signals. Extending the HOUND differentiator to handle vector-valued signals or signals in higher dimensions would be a valuable contribution.

Detailed Comments:

1. Mathematical Derivations:
   The mathematical derivations in the article are thorough and well-presented. The authors have provided detailed proofs for the convergence of the differentiator and the boundedness of the error. However, some of the derivations, particularly those involving the recurrence relations for the coefficients \(a_{m,d,n}\) and \(b_{d,n}\), could benefit from further clarification. While the authors claim that these formulas have been verified for a wide range of values, a formal proof of their correctness for arbitrary \(n\) would strengthen the theoretical foundation of the method.

2. Discretization Method:
   The proposed discretization method is innovative and well-suited for real-time applications. However, the article does not discuss the stability of the discretized system in detail. It would be beneficial to analyze the stability of the discretized differentiator, especially when dealing with irregular time steps or noisy signals.

3. Comparison with Existing Methods:
   The comparative analysis with super-twisting and high-gain differentiators is a strong point of the article. However, the authors could expand this comparison to include other state-of-the-art methods in numerical differentiation, such as Kalman filters or wavelet-based differentiators. This would provide a more comprehensive evaluation of HOUND's performance relative to the broader landscape of numerical differentiation techniques.

4. Parameter-Free Nature:
   The parameter-free nature of HOUND is one of its most interesting features. However, the article does not discuss the potential trade-offs associated with this approach. For instance, while the method does not require tuning, it may be less flexible in adapting to specific signal characteristics compared to parameterized methods. A discussion of these trade-offs would provide a more balanced view of the method's strengths and limitations.

5. Future Work:
   The article outlines several promising directions for future research, including the exploration of different discretization methods, the impact of noise variance, and the parallelization of the differentiation process. These are valuable suggestions, and pursuing these directions could further enhance the applicability and efficiency of the HOUND differentiator.

6. Figures:
The author should edit the figures so that they are easy to read in black and white, also some curves on the graphs completely overlap others and are not visible.

Conclusion:

The article presents a significant contribution to the field of numerical differentiation with the introduction of the HOUND differentiator. The method's parameter-free realization, combined with its ability to handle high-order derivatives and noisy signals, makes it a powerful tool for a wide range of applications. The theoretical analysis is rigorous, and the proposed discretization method is practical and efficient. However, the article could benefit from a more detailed discussion of computational complexity, empirical validation on real-world datasets, and an analysis of the method's stability and transient behavior. Overall, the HOUND differentiator is a promising advancement in numerical differentiation, and the article provides a solid foundation for future research in this area.

Recommendation:

I recommend the article for publication, provided that the authors address the aforementioned weaknesses and areas for improvement. Specifically, the authors should consider adding a section on computational complexity, expanding the empirical validation to include real-world datasets, and providing a more detailed analysis of the method's stability and transient behavior. Additionally, a broader comparison with other state-of-the-art methods would further strengthen the article's contribution to the field. With these improvements, the article will be a valuable addition to the literature on numerical differentiation.

---

### Official Review · Reviewer_hScC · 2025-02-27
**This paper introduces HOUND (High-Order Universal Numerical Differentiator), a new method for numerically differentiating signals up to an arbitrarily high order. The core idea is to represent the differentiator as a system of nonlinear differential equations that estimates a signal and all its derivatives (up to a chosen order) simultaneously. The authors derive an explicit closed-form solution for this system. Importantly, HOUND does not require manual tuning of parameters (such as gain or noise thresholds) specific to the signal’s characteristics. They also propose a discrete-time implementation of HOUND as a cumulative smoothing algorithm that operates online (streaming data) and can perform both interpolation and extrapolation of the signal without fitting any polynomial coefficients offline.**

**Rating:** 8
**Confidence:** 4

**Review:**

Strengths (Pros):
- The paper introduces a genuinely novel method (HOUND) for high-order differentiation that isn’t limited to first or second derivative – it can estimate many derivatives simultaneously, which is valuable for tasks requiring higher-order signal analysis.
- HOUND does not require tuning of gains or knowledge of noise bounds (Lipschitz constants). This universality and ease-of-use is a major advantage over traditional differentiators, making the method broadly applicable without laborious calibration.
- The authors provide strong theoretical underpinnings. They derive an explicit solution to the differentiator’s equations and prove error convergence properties for polynomial signals and bounded-error stability for general signals. This analytical depth adds credibility and insight into why the method works.
- Empirical results (simulations on a noisy polynomial signal and a harmonic signal with noise) demonstrate excellent performance. HOUND achieves high accuracy in estimating both the signal and its derivatives, outperforming classic high-gain and super-twisting differentiators in these tests. Notably, it maintains low error even near the boundaries of the data range, enabling effective extrapolation (whereas the compared methods’ accuracy degrades at the edges).
- The method inherently filters out high-frequency noise (due to the cumulative smoothing aspect). The experiments show that HOUND’s error continues to decrease over time (for the polynomial example, the error eventually becomes lower than the competing methods which plateau due to noise), indicating strong noise-rejection capabilities. Higher-order derivatives, which are usually very noise-sensitive, are actually estimated more accurately with HOUND as time progresses – a counterintuitive but valuable property.
- The proposed algorithm operates in an online fashion, processing data as it comes in without needing to store large windows. This makes it suitable for real-time applications and big data streams. It also can interpolate missing data and extrapolate beyond observed intervals on the fly, which is quite powerful for tasks like forecasting or filling gaps in time-series.

Limitations (Cons):
- The strongest theoretical results (zero asymptotic error) hold for polynomial signals of a certain order. Real-world signals might not be well-approximated by a polynomial of fixed order over long periods. If the signal deviates significantly from the polynomial model or has non-smooth behavior (e.g., discontinuities or abrupt changes beyond the assumed order), HOUND’s error will not vanish and could be larger than expected (though still bounded if the signal is differentiable and satisfies the Lipschitz condition). In practice, the user must choose the differentiator order n high enough to capture the signal’s complexity; if n is too low, there will be a bias (error floor) in the estimates.
- As noted in the paper, HOUND exhibits a transient phase where the estimates go through a distinctly nonlinear adjustment. There is a “sharp rise and fall” in the estimated values of derivatives at the start, with error sign changes at each step, before stabilizing. This means that early in the estimation process (or immediately after a sudden change in the signal), the outputs might overshoot or oscillate briefly. This behavior could be problematic in real-time control systems if not accounted for, since it could introduce momentary instability or require a “settling time” before the estimates become reliable.
- The continuous formulation uses terms like $1/t^m$ which become very large as $t$ approaches zero. In implementation, one cannot start at exactly $t=0$, and discretization must be handled carefully to avoid numerical issues at the beginning of the data stream. The paper uses a small time-step (e.g., $\Delta t = 0.0001$ in simulations) to mitigate this. However, extremely small time steps increase computational load and might not be feasible in all real-time systems. The paper doesn’t thoroughly discuss the sensitivity to the choice of sampling rate or how to initialize the differentiator safely when $t$ is small, which could be a practical limitation.
- Although conceptually straightforward (essentially integrating a chain of $n$ first-order ODEs), using a very high order $n$ increases the number of state variables and computations per time step. If one needed, say, 10th or 20th order derivatives, the algorithm would involve integrating a large system of equations and could become computationally heavy or more prone to numerical error accumulation. The paper does not provide an analysis of how large $n$ can be before running into performance issues or diminishing returns, which could be important for applications that might tempted to set $n$ very high.
- The experiments compare HOUND to two methods (a high-gain observer and a super-twisting differentiator), which are appropriate baselines from control theory. However, there are other numerical differentiation approaches (e.g., Savitzky–Golay filters, spectral methods, or machine learning regressors for derivatives) that were not compared. It’s unclear how HOUND stacks up against those in terms of accuracy or adaptability. A broader comparison would strengthen claims of universality. Also, the examples are relatively controlled (polynomial and harmonic signals with added noise); performance on more complex or chaotic real-world signals remains to be demonstrated.
- While not a flaw in the work per se, the paper’s heavy mathematical nature might make it less accessible to readers outside the specific communities of control theory or mathematical signal processing. The reliance on appendices for proofs and the lack of a high-level pseudocode could deter some practitioners. This is a minor presentation limitation that could be improved to widen the impact of the work.

---

### Decision · Program_Chairs · 2025-03-08

**Decision:**

Accept (Oral)

**Comment:**

Your article has been accepted and you can make a presentation on the article. All articles will be sorted by rating and within the available conference places one author from each article will be invited. If there are not enough places, then you will either have the opportunity to present remotely or come at your own expense!